# Movements of a Specialist Butterfly in Relation to Mowing Management of Its Habitat Patches

**DOI:** 10.3390/biology12030344

**Published:** 2023-02-21

**Authors:** Miloš Popović, Piotr Nowicki

**Affiliations:** 1Department of Biology and Ecology, Faculty of Sciences and Mathematics, University of Niš, Višegradska 33, 18000 Niš, Serbia; 2Institute of Environmental Sciences, Faculty of Biology, Jagiellonian University, Gronostajowa 7, 30-387 Kraków, Poland

**Keywords:** *Phengaris* (*Maculinea*) *teleius*, scarce large blue, dispersal, population size, displacement

## Abstract

**Simple Summary:**

European grasslands and their rich biodiversity have been shaped by humans for centuries through grazing and mowing. While mowing is perceived as good practice for sustaining meadows and conserving butterflies in the long term, the mowing event itself may have negative short-term effect on butterflies. A mosaic of different mowing regimes allowed us to explore this effect while studying movements of a wet meadow specialist butterfly, the scarce large blue. The main results showed that mowing negatively affected butterfly population size and increased butterfly dispersal probability. However, the increased dispersal led only to short-distance movements and was apparently undertaken by individuals not adapted well enough to emigrate. In turn, a larger area of habitat patch was found to be beneficial for promoting long-distance dispersal and hosted bigger butterfly populations. This means that large and unmown meadows support more viable populations and provide better connection among local populations. They are particularly important for persistence of the studied butterfly in a fragmented landscape. Thus, conservation programmes should aim to retain enough unmown habitat each year and to preserve large and interconnected habitat fragments.

**Abstract:**

Over the centuries, mowing and grazing have been crucial for sustaining populations of grassland insects and their overall diversity in Europe. While long-term positive effects of mowing have been studied in more detail, little is known about the direct impacts of mowing on adult butterflies. Here, we explore how different habitat management (mown, recovered after mowing and unmown) affects movements and population estimates of the endangered specialist butterfly *Phengaris teleius*. The results showed higher dispersal probability from mown (22%) and recovered meadows (16%) than from the unmown ones (9%). However, mowing shortened the average dispersal distances (mown = 102 m, recovered = 198 m, unmown = 246 m) and reduced butterfly population size. In contrast, a larger area of the habitat patches promoted long-distance dispersal and sustained larger populations. We hypothesise that mowing caused depletion of resources and triggered dispersal of poorly adapted individuals. This behaviour is maladaptive and could lead to higher dispersal-related mortality; thus, mowing should be avoided before and during the butterfly flight period. This study suggests that the species’ persistence in a fragmented landscape depends on large, unmown and interconnected habitats that support more viable populations, promote long-distance dispersal, and enable (re)colonisation of vacant patches.

## 1. Introduction

Europe has a long history of agriculture, accompanied by an ever-increasing impact on natural and semi-natural grasslands. In recent decades two opposite processes have been recognised as major threats to overall biodiversity: intensification of production in the developed and more accessible areas, and abandonment of traditionally managed agricultural land on the other side of the spectrum [1]. While habitat overgrowing could be reversed, agricultural intensification in grassland areas leads to using a large amount of pesticides and fertilisers that are known for their prolonged negative impact. The pronounced long-term effect of nitrogen deposition is recognised as one of the worst practices which have strong influence on specialist species of once rich natural grasslands [2,3,4]. The changes in European grasslands are known to affect a wide array of arthropod taxa [5], although direct effects of mowing seem to be more pronounced for species living within plants in contrast to ground-dwelling species, and for those with wings in contrast to wingless taxa [6,7]. Pollinators also show rapid responses to changes in grassland management, where inappropriate practice could have huge consequences [8,9].

Since most European butterflies are primarily grassland species, they show particularly rapid reactions to these habitat changes and steep declining trends [10]. The impacts of inappropriate habitat management practices, namely either abandonment or intensive use, are especially pronounced in specialist butterflies [11,12]. Among numerous examples, agricultural intensification led to the extinction of *Colias myrmidone* (Esper, 1781) in the Czech Republic and most of Europe [13] and of *Phengaris nausithous* (Bergsträsser, 1779) and *P. teleius* (Bergsträsser, 1779) in the Netherlands [14], while abandonment was the cause for extreme decline of *Chazara briseis* (Linnaeus, 1764) in Central Europe [15] and the extinction of *Phengaris arion* (Linnaeus, 1758) in Britain [16]. In the face of changing climate, past meadow management practices that were applied for centuries may not be enough to sustain butterfly diversity [17,18,19]. Unpredictable joint effects of climate change and other factors are also expected in the future. It has been shown that warming climate and nitrogen deposition jointly produce colder spring microclimates that negatively affect the populations of butterfly species hibernating as eggs or larvae [20]. To tackle the problem of butterfly decline, field practice generally suggests application of mosaic mowing regimes that resemble traditional land use, with low intensity of grazing and light mowing being preferred [21].

Mowing is known to influence populations of butterfly species [13,22,23] and their overall diversity [24,25,26]. In the short term, mowing produces a series of negative effects, with direct mortality caused by various machinery used in the process and indirect impact through the reduction of available host plants for caterpillars and nectar sources for adult butterflies or changing the habitat microclimate [27,28,29]. In contrast, in the long term, mowing saves meadow habitats from overgrowing, prevents the succession of vegetation, promotes plant species diversity and provides young host plant shoots required by some caterpillars [21,30,31,32,33]. Thus, the overall long-term effects of mowing are highly positive for grassland butterflies. However, this is true only if we properly adjust the timing and frequency of mowing to minimise the aforementioned negative short-term effects. From the perspective of conservation, late season mowing seems to be beneficial for overall butterfly diversity [25,34], while biannual mowing practice must be avoided [13,22,23,35]. In addition, mowing should be skipped periodically (rotational mowing), and some strips of grassland should be left unmown [30,35,36].

The long-term effects of mowing on butterfly populations are important in the context of conservation and have been frequently studied (see review in [30]). The same is true for the impacts of mowing on larval stages, being the most vulnerable period of the butterfly life cycle. On the other hand, direct effects on adults tend to be disregarded and remain mostly unknown. Little is known about the potential role of mowing as a driver of dispersal triggered by resource depletion, even if it can be predicted based on theoretical rationale. While meadow vegetation can recover a few weeks after a mowing event, its structure has been changed and resource availability reduced, enhancing intra-specific competition and affecting butterfly behaviour, such as oviposition, nectaring or movements. In this paper we analyse the short-term effects of mowing on adult *P. teleius* butterfly movements and their local population sizes. We hypothesise that mowing could cause resource depletion that promotes butterfly emergent dispersal (defined as movements between habitat patches) and increases individual displacement (defined as within-patch movements). In addition, we expect lower individual survival and smaller population size caused by direct mortality of individuals and their emigration.

## 2. Materials and Methods

### 2.1. Study Species

Large blues of the genus *Phengaris* (=*Maculinea*) are one of the best-studied butterflies, owning their popularity to their intriguing and complex life cycle, but also to their threatened status, with extinctions recorded across the European part of their range, followed by a few successful reintroductions [37,38,39]. They are recognised as flagship and iconic butterflies, with high priority in nature conservation. The focal species of the present study was the scarce large blue, *Phengaris teleius* (Bergsträsser, 1779), which shares all the above-mentioned characteristics as its wet meadow habitats have been rapidly shrinking across Europe in recent decades. The species is assessed as vulnerable in Europe [40], threatened in many countries and endangered in Serbia [41]. Its protection is based on the Habitats Directive within the European Union and the Bern Convention in other countries signing this international agreement.

*Phengaris teleius* is a Palearctic species found in humid meadows from the Pyrenees through Central Europe and Asia to Japan [42]. Great burnet (*Sanguisorba officinalis* L.) serves as the only larval host plant. The inflorescences of this plant are consumed by the caterpillars until the fourth larval stage, and further development continues only if the larva is adopted by specific ant species from the genus *Myrmica* [43,44]. Adopted larvae switch to feeding on ant brood and pupate within the ant nests, emerging as adults after one or two years [45]. Adults fly from July to the end of August. Numerous studies have investigated metapopulation structure and movements of *P. teleius* [46,47]. The species was found to form closed local populations occupying discrete host plant patches and to be relatively sedentary, with a short adult life span of about three days [48].

### 2.2. Study Area

Occurrence of *P. teleius* in the Republic of Serbia is limited to about 210 ha of wet meadows in the very north of the country. More details on this habitat and butterfly population are published elsewhere [49]. A complete study system includes three geographic areas (Subotica Sandland, Ludaš Lake and Selevenj Heath), which are delineated in five sampling localities in our studies, primarily for logistic reasons (Figure 1: letters A–E). These geographic areas are:

Subotica Sandland (46.169° N; 19.713° E; Figure 1: locality A), which is a sandy, dry area with meadows regularly surrounded by lines of trees and fragmented forests. The butterfly habitats are found within wet *Molinia* meadows developing at the lowest elevations and supported by the high level of underground waters sustained by the nearby Kireš River. With increased intake of water for agriculture and urban development, both from Kireš River and the underground reservoirs, these habitats are facing major threats in the future.Ludaš Lake (46.103° N; 19.801° E; Figure 1: localities B and C), which is a mosaic of urban, agricultural and wetland areas. The butterfly habitats are composed of wet meadows dominated by *Phragmites* or *Molinia* and located close to the nearby Kireš River or Ludaš Lake. These habitat patches are more fragmented due to intensive agriculture, with populations on small locality C experiencing strong fluctuations due to changing management practices on mostly privately owned land (authors’ personal observations).Selevenj Heath (46.138° N; 19.907° E; Figure 1: localities D and E), which is a large grassland composed of wet and saline meadows, interspersed with agricultural fields and swamps. Suitable habitats for *P. teleius* are dominated by both *Phragmites* and *Molinia*. These meadows are more continuous, and their portions are mown at different seasons, providing mosaic management practice crucial for conserving butterfly population and overall rich biodiversity of the meadows.

For the purposes of the present study, we sampled 46 habitat patches with a total size of 81 ha, with individual patches ranging from 64 m^2^ to 10 ha (median: 0.9 ha). The habitat patches (or simply patches) were defined as isolated meadow fragments covered with *S. officinalis* host plant and separated by roads, verges or different mowing regimes (Figure 1). All the patches were mapped in the field using GPS devices and their boundaries were subsequently drawn in QGIS guided by satellite imagery from Google Earth [50] and Landsat NDVI [51] to control for potential errors in delineating patch borders and time of mowing, respectively. It should be noted that only the part of the meadow containing suitable habitat and butterfly host plant was considered as habitat patch, and not the cadastral meadow area.

### 2.3. Data Collecting

Movement of butterflies was studied using mark–release–recapture (MRR) methods. The study was conducted by four persons from 19 July to 28 August 2014. We excluded the days with rain and low morning temperatures that could affect butterfly activities. Thus, the sampling was carried out every day, with a maximal interval between sampling occasions of three days during the bad weather. Capture sessions started in the morning, roughly at the same time for each person and the study locality. The duration of the sessions varied to keep the capture probability relatively constant. Consequently, it lasted longer in the peak of *P. teleius* flight period. Overall, during each session, we tried to cover most of the area within the habitat patches and to sample habitat patches with similar intensity. Unique individual numbers were written on the hind wing underside of each butterfly using a permanent (waterproof) marker pen (Figure 2a). Each butterfly was immediately released at the place of capture, low to the ground and with as little disturbance as possible. The exact coordinates were taken using a GPS device. The complete dataset and R script used to fit the models is available as Appendix A.

### 2.4. Statistical Analysis

We explored the differences in dispersal (dispersal probability and dispersal distances), displacement distances, survival, capture probability and population size among wet meadows with different mowing regimes. Dispersal includes movements of individuals from one breeding site (habitat patch) to another and was measured here through dispersal probability and dispersal distances. We defined dispersal (inter-patch movement) as a movement event during which the individual butterfly changes habitat patch [52]. Dispersal (emigration) probability was defined as the proportion of individuals captured at a given patch and recaptured elsewhere among all recaptured individuals originating from the patch, i.e., including those remaining in the patch. Displacement represents a movement event of an individual within the same habitat patch and reflects how individuals behave during regular activities within their usual home range [53,54]. To explore the effect of mowing, we classified meadow management regimes into three categories: unmown, recovered and mown (Figure 2b–d). The unmown patches should provide full potential resources for the butterflies. The recovered patches were mown at least 15 days before the investigated flight period, allowing the host plants to reflower and thus provide sufficient resources for butterfly feeding and oviposition. In contrast, mown meadows were those at which mowing took place less than 15 days before the flight period, and those should provide no or very limited resources for the butterflies.

Survival (*ϕ*) and capture probability (*p*) were estimated using the Cormack–Jolly–Seber (CJS) model [55] for each habitat patch with a large enough sample of (re)captures. The CJS model was fitted in *marked* package [56] in R statistical software [57]. We allowed survival and capture probability to vary between sexes but not among capture sessions and selected the best model using the Akaike information criterion (AIC) value, i.e., the model with the lowest AIC [58]. Total population size (N_total_) is calculated as the sum of recruitments (B’_i_), corrected for individuals that emerge and shortly afterwards die between consecutive capture days, as in Nowicki et al. [48].

The predictor variables included sex, patch area in m^2^ and connectivity. The area of each patch and other spatial data used to calculate connectivity were obtained from the patch polygons using R statistical packages *rgdal* and *rgeos* [59,60]. Patch connectivity was calculated using the well-established standard formula [52,61,62], in which connectivity of patch *j* is estimated versus *k* nearby patches within 3 km as shown in Equation (1):(1)$$S_{j}=\sum_{k\neq j}\exp\left(-{\alpha d}_{\mathrm{jk}}\right)A_{k}^{\xi }$$

Here, d_jk_ represents the distance between patches in km, *A_k_* is the area in m^2^, α is a measure of dispersal ability (1/average dispersal distance in km), and *ξ* is the scaling factor of immigration that was set to a constant value of 0.5 as suggested by Brückmann et al. [63]. The threshold value of 3 km defining the nearby buffer zone should be considered as conservative, since dispersal distance in our study system is usually below 1 km, the maximal observed distance is 1.8 km, and the estimated probability to disperse more than 3 km is less than 0.1% [64].

We explored the differences in dispersal probability among mowing regimes by fitting the binomial family of generalised linear mixed models (GLMMs). Distances of dispersal and displacement were calculated from geographical coordinates of butterfly captures with the *rgeos* package in R and compared using the gamma family of GLMMs. Mowing regime, butterfly sex, patch area, estimated population size and patch connectivity were used as independent variables (=predictors) in the analyses of movement distances.

In the second set of models, we explored differences in demographic estimates (survival, capture probability and population size) among habitat patches. Differences in survival and capture probability among habitat patches were analysed using the linear mixed model (LME) of *lme4* package [65] in R, with mowing regime, butterfly sex, patch area and connectivity adopted as independent variables. Differences in population size among habitat patches were compared using the Poisson family of GLMMs of the *lme4* package, with mowing regime, patch area and connectivity used as independent variables.

In all the models, we included locality as a random effect defining five separate geographical sites to account for potential differences among localities and persons conducting the study. Continuous variables used in the models were normalised by their mean and standard deviation using the *scale* function in R prior to the analyses. Model selection was based on AIC [58]. Typically, the model with the lowest AIC was chosen, but those with ΔAIC < 2 were occasionally also selected if they were less complex (i.e., had lower number of parameters) than the model with the lowest AIC.

## 3. Results

In total, we captured 3945 individuals, with 1228 of them being recaptured at least once during the study (1583 recaptures in total). Average survival (*ϕ*) and capture probability (*p*) were estimated using a CJS model for 32 habitat patches at *ϕ* = 0.79 ± 0.09, *p* = 0.32 ± 0.14, while the local population sizes varied between 102 and 453 adults. The details of the model selection procedure are given in Appendix A.

The best model describing dispersal probability included all the predictor variables and the interaction of population size, patch area and patch connectivity (Table 1). The model showed that dispersal rate was greatest from the mown meadows (0.22 ± 0.16), followed by recovered meadows (0.16 ± 0.06) and the unmown meadows (0.09 ± 0.04) (Figure 3a). Intersexual differences were the most evident factor, with average dispersal probability for females (0.20 ± 0.09) being significantly greater than for males (0.11 ± 0.05). The effects of patch area, connectivity and population size were more complex and not straightforward to interpret from the model. Nevertheless, fitting univariate models indicated that dispersal is higher from small butterfly populations (GLMM model: dispersal probability = −1.67 − 0.50 × population size), small patches (GLMM model: dispersal probability = −1.61 − 0.61 × patch area) and better-connected patches (GLMM model: dispersal probability = −1.37 − 0.16 × patch connectivity), with all these effects being statistically significant.

All the supported models that explained dispersal distances included the effect of mowing and sex, while other variables were shown to be of little importance (Appendix A). The best selected model (Table 2a, Figure 3b) predicted short dispersal distances of butterflies from recently mown meadows (102 ± 48 m), longer distances from recovered meadows (198 ± 72 m) and the longest distances when leaving the unmown patches (246 ± 88 m). Females’ dispersal distances were longer (189 ± 70 m) in contrast to those of males (146 ± 54 m). Increase in available patch area had only slight effect in promoting longer distance dispersal.

All the best models predicting butterfly displacement distances within habitat patches included the effect of sex (Table 2b, Figure 3c). Females (57 ± 8 m) were again shown to be more mobile in contrast to males (43 ± 6 m). Mowing did not affect butterfly displacement distances within the habitat patches. Population size, patch area and patch connectivity were less important predictors, but all had slight positive effects on displacement distance as shown by univariate models (GLMM models: displacement distance = 3.78 − 0.15 × patch area; displacement distance = 3.78 − 0.12 × population size; displacement distance = 3.77 − 0.07 × patch connectivity).

Survival and capture probability did not differ among mowing regimes and the best models selected by AIC included only the intercept term after accounting for the random effect of the locality. The best model explaining differences in population sizes among habitat patches included all the predictor variables as well as the interaction between patch size and connectivity. Unmown patches supported numbers of individuals approximately twice as high as mown or recovered patches (Table 3, Figure 3d). In addition, the local numbers of adults depended on the available area of the patch but were negatively affected by patch connectivity.

## 4. Discussion

### 4.1. Effects of Mowing

Mowing regime proved to be an important determinant of both dispersal of *P. teleius* butterflies and their numbers within habitat patches, while it had no apparent effect on displacement distances of individuals within the patch or the average survival and capture probability within the patches. We have found that the individuals from unmown patches rarely decide to disperse from the natal patch, but once the dispersal decision is made, they travel longer distances. It is well known that only a portion of butterflies within a population are adapted for dispersal, while most resident individuals perform only routine displacement movements [66,67]. On the other side, mowing triggers dispersal of more individuals, but consequently the dispersal distances are much shorter. A fine gradient of increasing dispersal probability could be observed from unmown through recovered and finally to mown meadows, suggesting that resource limitation, rather than the mowing event itself, is the likely proximate driver of dispersal. In other words, mowing probably causes resource depletion [27,29] and forces many individuals that are poor fliers and would otherwise stay within their natal patches to disperse. Similar results were obtained by experimentally decreasing habitat quality [68] or when butterfly population densities increased [69], and especially if this increase exceeded habitat carrying capacity [70]. However, we are not aware of any earlier studies documenting increased dispersal from habitat patch caused by mowing [27].

Although one could perceive the mowing-induced dispersal as a positive aspect for metapopulation functioning, the most important finding of this study suggests that such dispersal events, presumably undertaken by specimens poorly adapted to move through the matrix between habitat patches [47,70], most likely have no significant evolutionary benefit as they only lead to very short-distance movements. Although in our study system mown meadows were usually close to other suitable habitats, this is not always the case. More isolated butterfly populations could suffer from increased mortality during dispersal [46,71], and hence the effect of mowing will overall be negative for the species’ persistence in a fragmented landscape.

Females were found here to be the more mobile sex, in terms of dispersal probability, dispersal distances and displacement distances, which is concordant with the outcomes of earlier studies [54,72]. Males of *P. teleius* tend to form small home ranges, while females travel longer distances to find new host plants for oviposition and are more likely to disperse to new habitats. From a behavioural perspective, females of *P. teleius* perform short-duration but frequent flights and devote more time to rest and oviposition, while males spend more time flying to explore their surroundings for receptive females [73]. Accordingly, male dispersal is typically promoted by the scarcity of females, while female dispersal is promoted by the lack of resources for oviposition [74]. This could easily explain the fact that, in our study, female dispersal increased more pronouncedly after mowing, which diminishes the availability of oviposition sites.

Local populations in the unmown meadow patches were shown to be more numerous compared to the recovered or mown ones. This provides a strong argument against mowing shortly before or during the butterfly flight period, as already confirmed in various studies for *P. teleius* [22,23,35]. It is noteworthy that the habitat patches in Serbia are not managed specifically for the focal butterfly species. They rather provide a casual mosaic of small patches mown in different periods, which ensures the species survival in the long run through a relatively healthy metapopulation structure [49]. While such management is proven to be optimal for the species, some care must be taken to keep the mowing plots small and enhance diversity of mowing regimes, since a few years of inappropriate management on a large scale could easily push this species to regional extinction [75,76]. It should be noted that mowing is a relatively new practice. After the end of the recent ice age, forests dominated the European landscape, while open patches were sustained by abiotic factors or herbivores. Humans began to graze large animals ca. 5000 BC and cleaned vegetation by burning, whereas mowing was introduced afterwards but intensified and accompanied by fertilisers only in the last 60 years [77]. Thus, most European grasslands are biodiversity-rich semi-natural habitats with patch layout shaped by farming over the last millennia. However, these habitats are facing strong threats in the last hundred years and the conservation of their communities is an urgent matter that must involve both conservation managers and livestock farmers [14,77].

### 4.2. Effects of Habitat Spatial Structure

Besides the aforementioned effects of mowing regime and butterfly sex on movement and population estimates of *P. teleius*, the spatial parameters defining patch area and connectivity were also shown to be important. According to the metapopulation theory framework, it is widely accepted that dispersal is less likely from large habitat patches [52]. Smaller patches have larger patch circumference to patch area ratio, and therefore individuals more frequently encounter boundaries when moving around. Butterfly dispersal is initiated after crossing such a boundary and followed by long and straight flight in the matrix [78,79]. The effect of patch area on butterfly population size and within-patch displacement distances is logical and easy to grasp. However, it remains less clear how natal patch area could increase inter-patch dispersal distances of the butterflies. The potential explanation could lie in the fact that while at small patches all individuals are equally likely to encounter patch boundaries, at larger patches it happens much more frequently to better-flying individuals, which may subsequently be able to cover longer distances through the matrix. In summary, larger patches not only offer bigger source populations of *P. teleius* for colonisation of vacant patches and more viable genetic structure [16,38,80,81], they could also support emigration of individuals that are more adapted to dispersal.

As expected from the theoretical background, patch connectivity had a positive effect on emigration probability [52], while it negatively affected population size. The connectivity–dispersal relation is likely to be an adaptive trait, where strong selection acts against individuals from more isolated habitat fragments [46,66,67,68]. In turn, the negative influence of connectivity on local population size may appear counter-intuitive. However, this is probably the outcome of the spatial arrangement of patches within our study system, with a plethora of tiny and well-interconnected patches hosting small populations.

## 5. Conclusions

It must be stressed that this study covers a single specialist butterfly species, but to our knowledge, it also provides the first direct evidence that mowing promotes dispersal of a butterfly species from its natal habitat patch. To study this problem in more detail, one would require exceptionally detailed experimental design and ideally cover several butterfly species. Mowing is likely to cause resource depletion, both at the most recently mown meadows and at previously mown, now recovering meadows. This induced emergent dispersal but significantly decreases dispersal distances of *P. teleius* individuals. It may actually force poorly adapted individuals that would otherwise stay within their natal patches to disperse and elevate dispersal-related mortality. Therefore, we conclude that the effect of mowing on butterfly movements is predominantly disadvantageous, and the same is true for its effect on population size. From the perspective of conservation, mowing should be strictly avoided shortly before and during the butterfly flight period, while leaving some areas unmown is desirable.

Area size of the habitat patch is another important factor affecting populations of *P. teleius*. Although small patch area is well known to increase dispersal probability, the populations found in large meadows were significantly bigger and characterised by longer dispersal and displacement distances of the studied species. Consequently, large habitat patches ensure viable source populations, which are vital for colonisation of vacant habitat patches and metapopulation persistence in the long term. Therefore, conservation programmes should primarily preserve large and good-quality habitat fragments.

## Figures and Tables

**Figure 1 biology-12-00344-f001:**
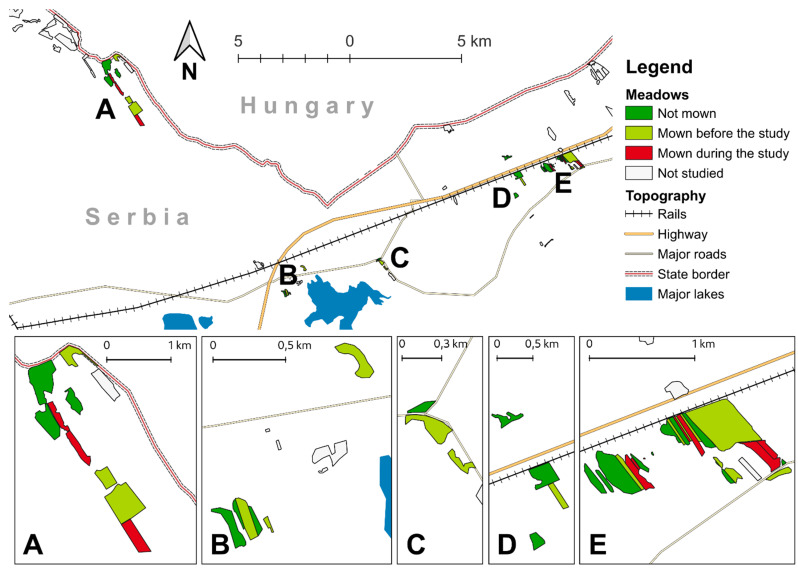
The map of the study region covered with mark–release–recapture (MRR) survey of the scarce large blue butterfly (*Phengaris teleius*). The meadow mowing regimes are shown in different colours. Meadows not included in the study had no butterflies in 2014 or were further away from the core study area and thus were not surveyed for logistic reasons.

**Figure 2 biology-12-00344-f002:**
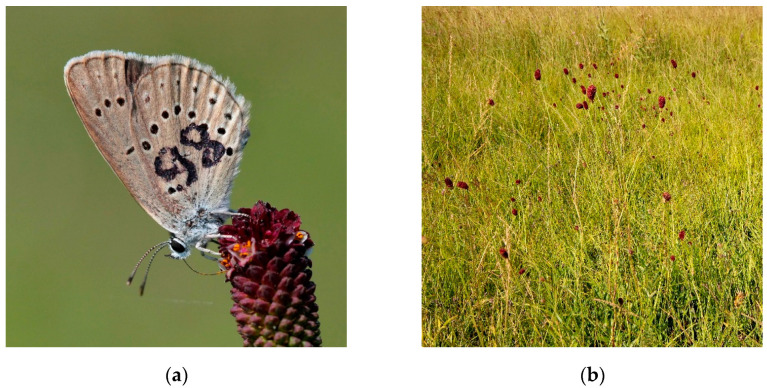
The study species and locality showing (**a**) marked scarce large blue butterfly (*Phengaris teleius*) and studied meadows classified in three categories as (**b**) unmown, (**c**) recovered and (**d**) mown.

**Figure 3 biology-12-00344-f003:**
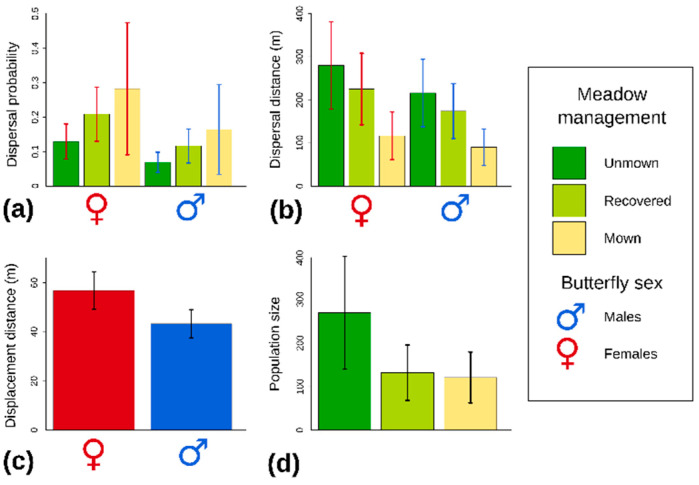
Results of the models predicting (**a**) the effect of mowing and butterfly sex on dispersal (emigration) probability and (**b**) dispersal distance, (**c**) the effect of butterfly sex on displacement distances and (**d**) the effect of mowing on population size of adult *Phengaris teleius* butterflies. The results are given as predicted responses from the fitted models with lines representing their SEs.

**Table 1 biology-12-00344-t001:** Dispersal (emigration) probability of *Phengaris teleius* estimated from the binomial family of generalised linear mixed models. The model included meadow mowing regime (mown, recovered and unmown), butterfly sex (males and females), patch connectivity, patch area, butterfly population size and the interaction of the last three variables.

Variable Name	Estimate ± SE	Z Value	*p* Value
Intercept	−1.907 ± 0.454	−4.200	**<0.001**
Mowing regime (recovered)	0.574 ± 0.172	3.346	**<0.001**
Mowing regime (mown)	0.974 ± 0.820	1.188	0.235
Butterfly sex (males)	−0.694 ± 0.127	−5.485	**<0.001**
Population size	−1.701 ± 0.229	−7.439	**<0.001**
Patch connectivity	0.994 ± 0.179	5.556	**<0.001**
Population size × patch connectivity	1.569 ± 0.346	4.534	**<0.001**
Population size × patch connectivity × patch area	0.890 ± 0.137	6.496	**<0.001**

Statistically significant *p* values are shown in bold.

**Table 2 biology-12-00344-t002:** Dispersal and displacement distances of *Phengaris teleius* estimated from the gamma family of generalised linear mixed models. The original model included meadow mowing regime (mown, recovered and unmown), butterfly sex (males and females), patch connectivity, patch area and butterfly population size.

Model	Variable Name	Estimate ± SE	T Value	*p* Value
**(a) Dispersal distance**	Intercept	5.632 ± 0.362	15.542	**<0.001**
	Mowing regime (recovered)	−0.217 ± 0.217	−2.563	**0.010**
	Mowing regime (mown)	−0.875 ± 0.308	−2.840	**0.004**
	Butterfly sex (males)	−0.258 ± 0.079	−3.287	**0.001**
	Patch area	0.090 ± 0.044	2.025	**0.043**
**(b) Displacement distance**	Intercept	4.040 ± 0.133	30.249	**<0.001**
	Butterfly sex (males)	−0.272 ± 0.050	−5.467	**<0.001**
	Population size	0.089 ± 0.028	3.118	**0.002**
	Population size × patch area	−0.301 ± 0.068	−4.435	**<0.001**

Statistically significant *p* values are shown in bold.

**Table 3 biology-12-00344-t003:** Population size of *Phengaris teleius* butterflies explored through the Poisson family of generalised linear mixed models. The model included meadow mowing regime (mown, recovered and unmown), patch connectivity and patch area as independent variables.

Variable Name	Estimate ± SE	Z Value	*p* Value
Intercept	5.596 ± 0.480	11.666	**<0.001**
Recovered	−0.712 ± 0.030	−23.657	**<0.001**
Mown	−0.799 ± 0.043	−18.665	**<0.001**
Patch connectivity	−0.117 ± 0.013	−9.003	**<0.001**
Patch area	1.692 ± 0.026	64.666	**<0.001**
Patch connectivity × patch area	−0.569 ± 0.012	−47.472	**<0.001**

Statistically significant *p* values are shown in bold.

## Data Availability

Data is contained within the Appendix A.

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
