# Peer review of "Movements of a Specialist Butterfly in Relation to Mowing Management of Its Habitat Patches"

_biology, 2023, doi:10.3390/biology12030344_

Round 1

Reviewer 1 Report

This is very useful paper, which finally - given the large number of studies of butterfly populations using mark-recapture, and similarly large studies of butterfly responses to interventions such as mowing - connects the two methodological approaches, using mark-recapture to investigate population-level responses of an endangered butterfly to mowing of cultural grasslands. 

The paper is well written, uses up-do-date analytical methodology, and I am convinced that it is based on much effort in field and innovative analytical approaches. Consequently, I highly recommend its publication. 

My comments, below, are only very minor suggestions, intended to increase the appeal of the paper. 

1) lines 51-54. I have some reservations regarding the claim that "butterflies are primarily grassland species". It certainly applies in Europe, and perhaps in other long-farmed temperate areas, but one can dispute this, e.g., in wet tropics, where many species dwell in tree canopy, and certainly in other biomes. Please, re-word.

2) Similarly, even in this section of Introduction of elsewhere, it may be worth noting that the whole issue with mowing as radical, but necessary, habitat disturbance, applies not only to butterflies, but to other grassland/meadow insect groups. I understand that finding an appropriate reference to hymenopterans, orthoperans or moths can be more difficult than in the case of butterflies, but such expansion of focus will make the paper less butterfly-centric. 

3) Though I absolutely agree that mowing maintains the habitats for species such as P. teleius, and in the same time affects its populations if practiced in wrong time / over too large areas, and hence support the notions of patchy mowing, mosaic-like management etc., we should all be aware that as an ecological factor, mowing is quite young thing, not older than Neolithic revolution (and probably much younger, as it requires technologically advanced iron tools!). So a mention or guess, perhaps to Discussion/Conclusion, regarding the deep-past dynamics of habitats of P. teleius, and ensuing habitat structure, may be useful. It appears that erratic grazing of alluvial locations by large grazers was the "precursor" of current mowing; resulting vegetation structure would likely be very notched and patchy, which may provide a guidance for future management of P. teleius habitats. 

Author Response

Dear reviewer,

Thank you for good suggestions for improving our paper and positive review of the manuscript. As all your suggestions are well in place, we tried to correct the introduction and discussion sections accordingly.

Specific reply to the comments:
1. Rephrased.
2. Yes, the introduction section was narrowed to the butterflies quite quickly! We have expanded it not to briephly mention the effects of grassland management (mowing) on other arthropod species and pollinators in general, which now seems to be very important for providing us with valuable ecosystem services. Actually, there seems to be quite a lot of specialised literature on the effect of grassland management for each insect group. We tried to focus on the studies covering diversity of several arthropod groups and linking grassland diversity with mowing management practice.
3. Good sugestion. The discussion section is now expanded to introduce the reader with short history of formation of European grasslands and, more importantly, to the very important link between grassland and farming, that is a key for their preservation and a reason for their destruction in the last 100 years. Note that the text is very general, not specific to P. teleius butterfly, but it should be a good introduction to the conservation practicioners.

King regards,

Miloš Popović

Reviewer 2 Report

Review for manuscript entitled “Movements of a specialist butterfly in relation to moving management of its habitat patches” (ISSN 2079-7737) submitted for publication to the journal Biology.

General Comments:

This manuscript describes the effects that mowing or not mowing have on the movements (emigration) of a threatened/endangered species of butterfly. I enjoyed reading the manuscript, as I found the research question interesting. I highly recommend that the authors consider asking for some grammatical assistance with the writing, but in general the writing is good. The research question is interesting and the methodology is good. One of my biggest concerns with the paper is that the authors use this study to make rather large generalizations about the effects of mowing on butterfly populations; however, this investigation only reports the results from one rare butterfly species. I highly recommend that the authors consider this aspect of the research and rewrite the general claims made in this paper to be more specific to this particular species. Given the purpose of the project it would have been better to consider multiple species of Lepidoptera that represent a range of population sizes (i.e., very common species to very rare species). Because this paper reports only one species of butterfly it should be written that way, without claims as to how mowing affects general butterfly population dynamics and rather only the population dynamics of one butterfly species.

I also highly recommend that the authors consider adding another figure that shows a picture of the butterfly (maybe one that also is marked with a pen for the study) and images of some of the study sites (maybe show an image of a mowed treatment, unmown, and recently mown. It would be good for the reader to see these things to help with understanding the problem presented in the manuscript.

Specific Comments:

Introduction –

Line 51, I would rewrite the sentence “Being primarily grassland species, butterflies….”. I wouldn’t consider butterflies as primarily grassland species (e.g., huge diversity of butterflies in tropical rainforests).

Line 81, please remove the phrase “…every now and then” and replace it with “periodically” or something similar.

Line 88, I believe the word “moving” should be the word “mowing”

Materials and Methods –

Line 165, I would prefer different wording and more accurate information to replace the phrase “…on fine weather days…”, as this is quite subjective.

Discussion –

line 331, please reword “…males spend more time on wings to…” to “..males spend more time flying…”

Author Response

Dear reviewer,

Thank you for the good and positive review of our manuscript. We have tried to correct the issues, and all the specific comments were accepted in the revised manuscript. It is good idea to include additional figures, especially with different mowing regimes. This will help the majority of readers, that are not familiar with Scarce Large Blue habitat, to understand the topic better. The language of the manuscript is checked by our colleague and should be significantly improved now.

The generalisation issue probably requires more detailed explanation. It is extremely hard to track multiple species using mark-release-recapture methodology and to study different mowing regimes without detailed experimental design. In the present study we were lucky to have a diversity of mowing regimes and large geographic coverage to ensure relatively good sample for the analyses. Other studies we know of were tracking the effects of mowing along the transect lines, thus covering more species, but not allowing the estimates of dispersal and other population parameters. With additional corrections to the text, we hope it is more clear that only one species is covered by our study. The only part that remains more general is the conslussion section, but our findings seems to fit the general theory. We just added some additional evidence for already recomended grassland management practices of butterfly habitats, but based on new knowledge on dispersal triggered by the mowing event. Since the emigraiton is promoted by resource depletion, it is reasonable to assume that other grassland are affected in similar manner.

Kind regards,

Miloš Popović